# Universality of Dicke superradiance in arrays of quantum emitters

Stuart J. Masson [1✉] & Ana Asenjo-Garcia [1✉]

Dicke superradiance is an example of emergence of macroscopic quantum coherence via correlated dissipation. Starting from an initially incoherent state, a collection of excited atoms synchronizes as they decay, generating a macroscopic dipole moment and emitting a short and intense pulse of light. While well understood in cavities, superradiance remains an open problem in extended systems due to the exponential growth of complexity with atom number. Here we show that Dicke superradiance is a universal phenomenon in ordered arrays. We present a theoretical framework – which circumvents the exponential complexity of the problem – that allows us to predict the critical distance beyond which Dicke super-radiance disappears. This critical distance is highly dependent on the dimensionality and atom number. Our predictions can be tested in state of the art experiments with arrays of neutral atoms, molecules, and solid-state emitters and pave the way towards understanding the role of many-body decay in quantum simulation, metrology, and lasing.

[1] Department of Physics, Columbia University, New York, NY 10027, United States. ✉email: s.j.masson@columbia.edu; ana.asenjo@columbia.edu

Atoms in close proximity alter each others' radiative environment and collectively interact with light[1–4]. The "environment" for each of the atoms depends on the internal state of all others, which changes in time. For fully-inverted atoms at a single spatial location, this leads to the emission of a short pulse of light that initially rises in intensity, in contrast to the exponential decay of independent atoms. This "superradiant burst", or Dicke superradiance, occurs because atoms synchronize as they decay, locking in phase and emitting at an increasing rate. Superradiant bursts have been observed in a variety of dense disordered systems[3–9]. Dicke superradiance has also been demonstrated in cavities[10,11], where the condition of atoms at a point is emulated by the confinement of the optical field to zero dimensions. In this high-symmetry scenario, atoms are indistinguishable from each other, and can only occupy states that obey a particle-exchange symmetry. This restricts the Hilbert space to permutationally symmetric states, whose number scales linearly with atom number, thus making the dynamical evolution exactly solvable.

Numerical studies of superradiant emission in extended geometries (of sizes larger than the emission wavelength) have been limited to small numbers of atoms[12,13], small numbers of excitations[14], or uniform atomic densities where specific atomic positions are not taken into account[15]. However, recent experimental demonstrations of ordered atomic arrays, via optical tweezers[16–21] and optical lattices[22–25], open a new world of possibilities, where hundreds of atoms can be placed in almost arbitrary positions. These setups thus demand a new outlook on the problem, which has remained open until now due to the exponential growth of the Hilbert space. In extended systems, particle-exchange symmetry is broken and numerical calculations require a Hilbert space which grows as $2^N$, where $N$ is the atom number.

Here, we introduce a theoretical framework that scales linearly with atom number and allows us to demonstrate that Dicke superradiant decay generically arises in extended systems, below a critical inter-atomic distance that depends on the dimensionality. We do so by noting that there is no need to compute the full dynamical evolution of the system: the nature of the decay can be deduced from the statistics of the first two emitted photons. We find that as the inter-atomic distance increases, there is a smooth crossover between a superradiant and a monotonically decreasing emission rate, as shown in Fig. 1. We obtain an analytical

"minimal condition" for Dicke superradiance, which is universal and provides a bound on the maximal inter-atomic separation to observe this phenomenon. This enables us to study the role of geometry in the decay of very large atomic arrays, a significant conceptual advance on a decades-old problem.

## Theory

We first present the theoretical toolbox to describe the dynamics of a collection of atoms interacting via a shared electromagnetic field. We consider $N$ identical two-level atoms of spontaneous emission rate $\Gamma_0$ and transition wavelength $\lambda_0$ placed in free space with arbitrary positions. After tracing out the electromagnetic field using a Born–Markov approximation[26,27], the atomic density matrix $\rho = |\psi\rangle\langle\psi|$ evolves as

$$\dot{\rho} = -\frac{i}{\hbar}[\mathcal{H}, \rho] + \underbrace{\sum_{\nu=1}^{N} \frac{\Gamma_\nu}{2}\left(2\hat{\mathcal{O}}_\nu \rho \hat{\mathcal{O}}_\nu^\dagger - \rho \hat{\mathcal{O}}_\nu^\dagger \hat{\mathcal{O}}_\nu - \hat{\mathcal{O}}_\nu^\dagger \hat{\mathcal{O}}_\nu \rho\right)}_{\text{dissipative evolution: correlated photon emission}},$$

(1)

where the Hamiltonian $\mathcal{H}$ describes coherent interactions between atoms and $\{\hat{\mathcal{O}}_\nu\}$ are operators that represent how atoms "jump" from the excited to the ground state by collectively emitting a photon. Jump operators are found as the eigenstates of the $N \times N$ dissipative interaction matrix $\mathbf{\Gamma}$ with elements $\Gamma^{ij}$, proportional to the propagator of the electromagnetic field (i.e., the Green's function) between pairs of atoms $i$ and $j$ (see refs. [12,26–28] and "Methods"). The corresponding eigenvalues provide the jump operator rates $\{\Gamma_\nu\}$, which represent how frequently such a jump occurs. Each of these jump operators imprints a phase in the atoms, and generates a photon with a specific spatial profile in the far field. They thus can be understood as collective "decay channels" for the atomic ensemble.

As we demonstrate below, Dicke superradiance is preserved as long as the number of (relevant) decay channels is small. This occurs because dissipative interactions (rather than coherent Hamiltonian dynamics) are responsible for the suppression of superradiance in ordered arrays[12,13]. In the paradigmatic example studied by Dicke, where all atoms are exactly at one point, only one of the decay channels is bright (with decay rate $\Gamma_{\text{bright}} = N\Gamma_0$), while all the others are completely dark (i.e., $\Gamma_{\nu \neq \text{bright}} = 0$). This means that the only possible decay path to the ground state for atoms that are initially excited is through repeated action of the bright operator, which imprints a phase pattern in the atoms that is reinforced in each process of photon emission. Coherence emerges via this dissipative mechanism, which leads to the development of a macroscopic dipole through synchronization and to a rapid release of energy in the form of a superradiant burst.

In ordered arrays, the number of bright decay channels can be controlled by the inter-atomic distance. In principle, all jump operators are allowed to act. For small lattice constants, their decay rates vary dramatically due to constructive and destructive interference. They can be larger (bright) or smaller (dark) than the single atom decay rate $\Gamma_0$. Extremely dark rates (which are strictly zero in the thermodynamic limit) emerge for inter-atomic separations below a certain distance that depends on the dimensionality of the array[29]. As the distance grows, the distribution of the decay rates becomes more uniform. This leads to a strong competition between different decay channels, and to decoherence through the randomization of the atomic phases after several emission processes have occurred.

We show here that Dicke superradiance generically occurs in arrays, but only below a critical inter-atomic distance, which can be calculated with a complexity that scales only linearly with system size. For a fixed atom number, the superradiant burst

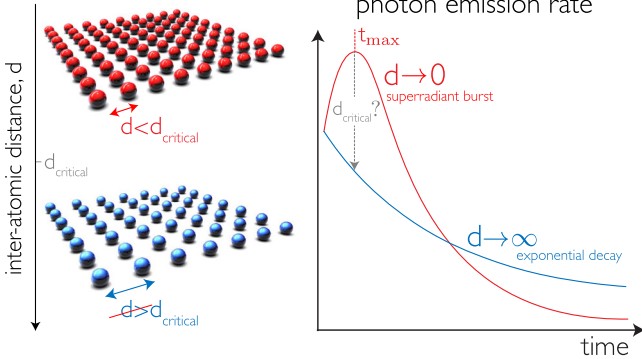

**Fig. 1 Many-body decay is determined by the distance between atoms and the array's dimensionality.** Inverted atoms placed at the same location ($d \to 0$) interact with each other and decay collectively via the emission of a burst of light, with a peak at time $t_{\text{max}}$. This is the hallmark of Dicke superradiance. In contrast, atoms that are far separated ($d \to \infty$) emit as single entities, in the form of an exponentially decaying pulse. For extended finite arrays, there is a critical distance at which the crossover between a superradiant burst and monotonically decreasing emission occurs.

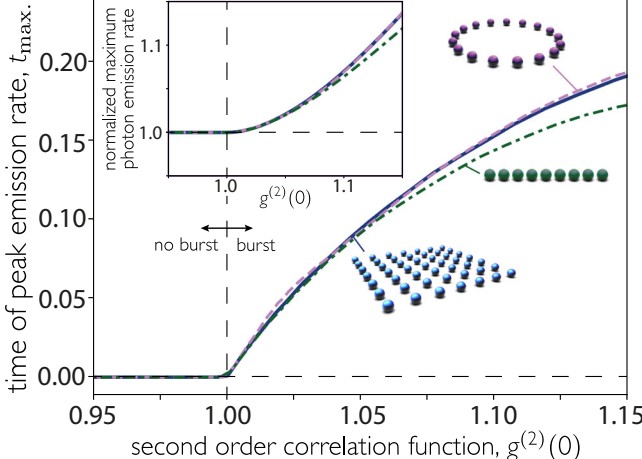

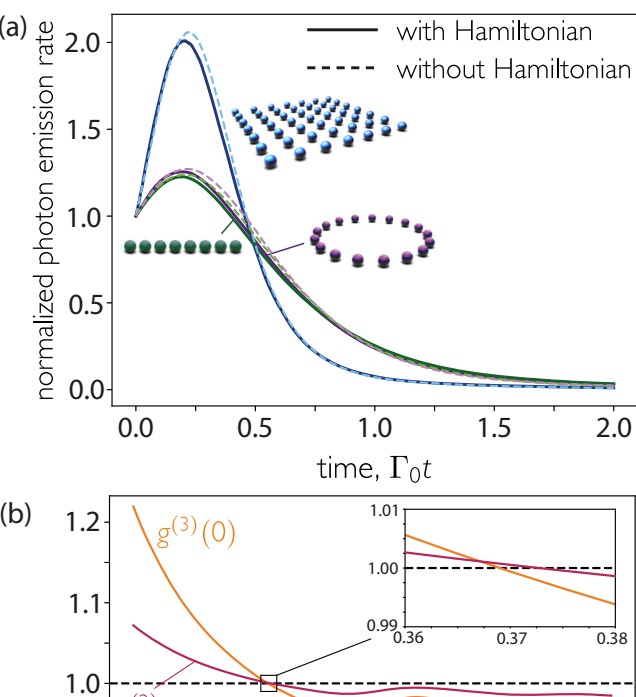

**Fig. 2 Photon statistics predicts Dicke superradiance.** We calculate $g^{(2)}(0)$ (at $t = 0$), as an enhanced two-photon emission rate is a pre-requisite for a burst. The time at which the photon rate is maximum ($t_{max}$) as a function of the second-order correlation function (at $t = 0$) shows that $t_{max} > 0$ only if $g^{(2)}(0) > 1$. Inset: Maximum intensity, normalized by intensity at $t = 0$. In both plots, all nine atoms are initially excited, with polarization perpendicular to the array.

diminishes as the inter-atomic distance increases, eventually being replaced by a monotonically decaying pulse. The crossover between these regimes is marked by an infinitesimally small burst that occurs at $t = 0$[13].

Our key insight is that atomic synchronization occurs immediately or not at all, and thus the nature of the decay can be characterized from early dynamics. In particular, one can predict the presence of a superradiant burst based solely on the statistics of the first two emitted photons. The minimum requirement for a superradiant burst to occur is that the first photon enhances the emission rate of the second. This is captured by the second-order correlation function

$$g^{(2)}(0) = \frac{\sum_{\nu,\mu=1}^{N} \Gamma_\nu \Gamma_\mu \left\langle \hat{\mathcal{O}}_\nu^\dagger \hat{\mathcal{O}}_\mu^\dagger \hat{\mathcal{O}}_\mu \hat{\mathcal{O}}_\nu \right\rangle}{\left( \sum_{\nu=1}^{N} \Gamma_\nu \left\langle \hat{\mathcal{O}}_\nu^\dagger \hat{\mathcal{O}}_\nu \right\rangle \right)^2}, \tag{2}$$

where the expectation value is taken at the initial state, i.e., $\left| \psi(t=0) \right\rangle = \left| e \right\rangle^{\otimes N}$. When this quantity is greater than unity, the decay is characterized as superradiant. Figure 2 shows the correlation between $g^{(2)}(0)$ and the presence or absence of a burst for small atom numbers, for which we can calculate the full dynamics. As soon as $g^{(2)}(0) > 1$, the time of maximum emission deviates from zero (i.e., the burst occurs at a finite time). Moreover, the second-order correlation function increases along with the height of the peak of the photon emission rate, and is below unity when the rate is peaked at $t = 0$.

By calculating $g^{(2)}(0)$ analytically (see "Methods"), we obtain the minimal condition for Dicke superradiance:

$$g^{(2)}(0) > 1 \quad \Leftrightarrow \quad \mathrm{Var}\left( \frac{\{\Gamma_\nu\}}{\Gamma_0} \right) > 1, \tag{3}$$

where Var is the variance of the decay rates of the jump operators. This expression is exact and universal, and does not involve any assumption about the atomic positions. Small inter-atomic distances maximize the variance of the decay rates, as most jump operators will be dark (with $\Gamma_\nu \simeq 0$) and just a small number of them will be bright (with a large $\Gamma_\nu$).

We note that the complexity of the problem has decreased tremendously: from solving a differential equation in an exponentially

growing Hilbert space to diagonalizing a matrix whose dimension scales linearly with atom number. This allows one to find the distance at which Dicke superradiance disappears in arbitrary geometries with an extremely large atom number, as all the necessary details are captured in the dissipative interaction matrix $\mathbf{\Gamma}$. Of course, one has to pay a price for this reduction in complexity. As we cannot calculate the full evolution, we can only predict whether a superradiant burst is going to occur or not. Extracting information about the height of the peak or the time at which it will appear requires a different approach[30,31].

To prove that the above inequality can be used to characterize Dicke superradiance, we demonstrate that the system does not rephase at later times, either through Hamiltonian action or through further dissipation. First, Hamiltonian dynamics are not significant at early times, as shown in Fig. 3a. Due to the ordered nature of the array, each atom (except those near the boundaries) experiences a similar environment and local dephasing due to Hamiltonian action is thus minimized. To further confirm this point, we consider a time delay between the emission of the first two photons, during which the Hamiltonian acts, and find the Hamiltonian adds a slow dephasing to the atoms but, importantly, does not enhance photon emission (see Supplementary Fig. 2). Second, dissipation into different channels cannot rephase the atoms, as the process is irreversible. With each photon that is emitted, there is one less atom able to emit. To obtain a

**Fig. 3 Role of coherent and dissipative evolution in dephasing and suppression of Dicke superradiance. a** The coherent evolution does not significantly modify the early time dynamics, thus preserving superradiance, as shown by the full evolution of the master equation (i.e., Eq. (1)) for 16 initially excited atoms with inter-atomic distance $d = 0.1\lambda_0$ arranged in different geometries with and without Hamiltonian interactions. Emission rate is normalized by that at $t = 0$. **b** Three-photon decay is never enhanced unless two-photon emission is too, as demonstrated by the second- and third-order correlation functions, plotted as a function of the inter-atomic separation for a square 2D array of 6 × 6 atoms. In all cases, atoms are polarized perpendicular to the array.

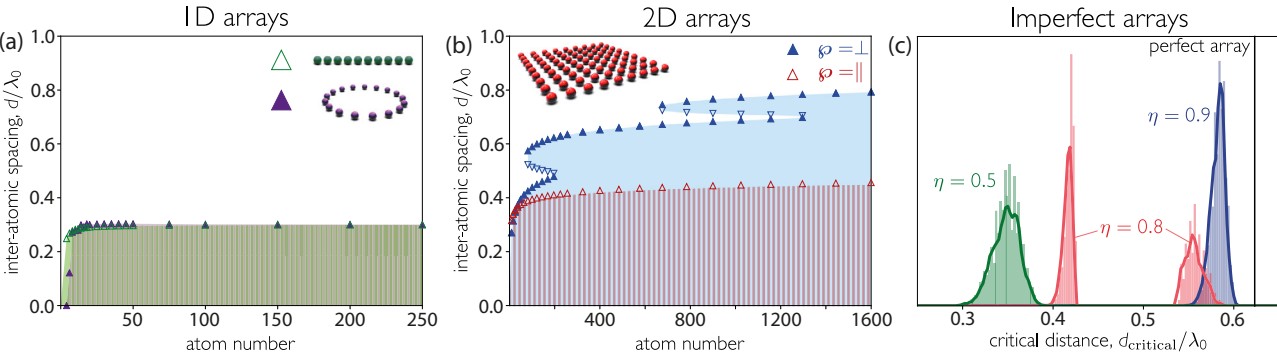

**Fig. 4 Dicke superradiance is universal and appears (below a critical distance) for arrays of any dimensionality, including imperfectly filled ones.**
**a**, **b** Boundaries between the burst (colored) and no-burst (white) regions as a function of inter-atomic distance $d$ and atom number for **a** chains and rings and **b** square arrays. The crossover occurs where $g^{(2)}(0) = 1$. Upward pointing and downward pointing triangles represent points where, with decreasing $d$, $g^{(2)}(0)$ goes above and below unity, respectively. **c** Critical distance for different filling fractions $\eta$. The histogram shows 2000 configurations of a $12 \times 12$ site square arrays stochastically filled with efficiency $\eta$. Envelopes are calculated as rolling averages. Atoms are polarized **a** parallel to the array (for the ring this implies a spatially dependent polarization), **b** perpendicular (blue) and parallel (red) to the plane, and **c** perpendicular to the plane.

superradiant burst, the induced atomic correlations that emerge through decay must be large enough to compensate for the loss of emitters, a process that gets harder the more photons have been radiated away. We characterize this through the third-order correlation function, which can be analytically calculated (see "Methods"). In all geometries considered here, we find that the third photon is never enhanced when the second photon is not. Therefore, further jumps do not rephase the array, as shown in Fig. 3b, where $g^{(3)}(0)$ drops below unity at a slightly smaller distance than $g^{(2)}(0)$. As anticipated, the second photon is always the last one to lose its stimulated enhancement.

## Results and discussion
Contrary to the accepted understanding in the literature[3,4], we find that large chains and rings behave almost identically, as both do not emit a superradiant burst above $d_{\mathrm{critical}} \approx 0.3\lambda_0$, as shown in Fig. 4a. Despite the ring's particle-exchange symmetry, the difference between the ring and the chain is negligible for large atom number. This is because dephasing is caused by competition between multiple decay channels, which exist regardless of the array topology[13]. Interactions across the diameter of the ring are very weak, so the exchange symmetry does not matter, as the atoms essentially see the same local environment in both cases.

Two- and three-dimensional arrays also display Dicke superradiance, at larger inter-atomic separations than those found in chains. Interestingly, the total size of the array is much larger than a wavelength. Figure 4b shows the critical distance for a two-dimensional (2D) square array of up to $40 \times 40$ atoms. In this geometry, the critical distance is not monotonic with the atom number. These sudden variations are due to "revivals" in $g^{(2)}(0)$, which can be seen in Fig. 3b, associated with changes in the distribution of $\{\Gamma_\nu\}$ as the lattice constant hits certain geometric resonances (see ref. [32] and Supplementary Fig. 1). For large array (of $N \sim 40 \times 40$ atoms), the critical distance is as large as $d_{\mathrm{critical}} \approx 0.8\lambda_0$ for atoms polarized perpendicular to the array surface, and it seems to continue increasing with atom number, albeit slowly.

Dicke superradiance is due to the dominance of particular decay channels, whose emission is enhanced due to constructive interference. Since the sum of the decay rates is always $N\Gamma_0$ (regardless of the atomic positions), these bright decay channels must be balanced by dark decay channels to maximize the variance. In ordered arrays, the presence of extremely dark channels is explained by energy–momentum mismatch, where some channels correspond to spin waves with wave-vectors outside the light cone[29,33]. In 2D arrays, the spin wave with equal phase on all

sites, with an in-plane wave-vector $\mathbf{k} = 0$, emits perpendicular to the array. If the atomic dipole axis points in that direction then emission is forbidden, creating a region of subradiance that persists up to $d < \lambda_0$[29]. Hence, the crossover between superradiant to monotonic decay occurs at much larger distances for atoms with this polarization. The same phenomenon exists in three-dimensional (3D) lattices for any linear polarization axis. Large 2D and 3D lattices both have values of $d_{\mathrm{critical}}$ well beyond $\lambda_0$[31,34,35]. For these higher dimensions, the dominance of certain channels is maintained due to robust constructive interference between many neighbors, compensated by large numbers of somewhat subradiant, but not perfectly dark, channels.

We demonstrate that Dicke superradiance is robust to imperfections typically found in experiments, such as filling fraction smaller than unity. Figure 4c shows the bound for stochastically generated $12 \times 12$ arrays filled with efficiency $\eta$. For $\eta = 90\%$, there is a small reduction in the critical distance. However, at $\eta = 50\%$, the drop is much larger. This is because the revivals in $g^{(2)}(0)$ are particularly muted by imperfect filling and, at this efficiency, do not breach unity. This phenomenon is also responsible for the splitting of the values of $d_{\mathrm{critical}}$ at 80% filling efficiency. Superradiance is also robust to position disorder and small imperfections in the initial state (see Supplementary Fig. 3).

Dicke superradiance should thus be observable in experiments with arrays of inter-atomic separation below the critical distance, which are close to being achieved in state-of-the-art setups[36,37]. It is important to notice that the critical distance does not signal a sharp transition between monotonic decay and superradiance, but instead a smooth crossover. Experimental signatures would be observable well below this bound. Besides atomic tweezer arrays and optical lattices, solid-state emitters hosted in bulk crystals[38,39] or in 2D materials[40–42] are good candidates to observe this physics. These systems can achieve small lattice constants, although they present other difficulties, such as inhomogeneous broadening and non-radiative decay. Nevertheless, Dicke superradiance is robust against these sources of imperfection (see Supplementary Fig. 4).

Superradiance in an extended array is very different from superradiance in a cavity. In the latter, superradiance involves three phenomena simultaneously: a growth in the photon emission rate, a rapid increase of the population of the cavity mode (due to the burst), and an $N^2$-scaling of the radiated intensity peak. These three concepts are not equivalent for extended arrays in free space, and this has experimental consequences. First, in free space, photons are scattered in all directions, and the relevant

geometry is not only that of the array, but that of the array together with the detector. In this work, we effectively integrate over all directions, which would correspond to collecting light over a large solid angle. As photon emission after a jump is directional[12,13], the burst is most optimally measured by a detector placed at the location where the far field distribution of the brightest jump operator is maximal. We note that our methods can be extended to account for "directional superradiance." Recent work[34] has shown that, unsurprisingly, the critical distance depends on the angular position of the detector. Second, the peak intensity may no longer scale as $N^2$. Finding the exact scaling is numerically challenging as it requires full dynamical evolution, though it should be accessible in experiments. Nevertheless, we speculate that the scaling will depend on the dimensionality and inter-atomic distance, and will be slower than $N^2$ (approaching $N$ for one dimension (1D) and with a power law whose exponent increases with dimension).

In conclusion, we have put forward a universal criterion that shines light into the physics of Dicke superradiance in extended systems. We have also demonstrated that Dicke superradiance universally appears in atomic arrays. We have bounded the critical distance that signals the crossover between monotonic decay and a superradiant burst, which is far larger than previously anticipated (for arrays of dimensionality higher than 1D). This bound is found by diagonalizing a matrix that scales only linearly with atom number. This method bypasses the exponentially growing Hilbert space required for full evolution by simplifying the problem to the statistics of the first two photons, which allows us to predict superradiance for very large arrays. Our approach could potentially be applied to disordered atomic ensembles[43,44] (where very small inter-atomic distances are achievable, but introduce large Hamiltonian frequency shifts that may need to be accounted for), to other types of Markovian electromagnetic reservoirs, such as nanophotonic structures[45,46] (by simply changing the Green's function[47]), and to emitters with more complex internal or hyperfine structure[48–50].

The understanding of many-body decay provided by our work is critical for developing robust and scalable quantum applications, ranging from quantum computing and simulation to metrology and lasing. In particular, our work is relevant for Rydberg atom quantum simulators[51–53], where Rydberg states may decay via long-wavelength transitions. These decay paths may be superradiantly enhanced at short distances[54,55]. Atomic arrays are also used in state-of-the-art atomic clocks and other precision measurement experiments[56,57]. As such systems shrink, it is crucial to understand the impact of collective dissipation. Finally, controlling the light emitted by an atomic array enables its use as an optical source. We have demonstrated that geometry can be used to alter the collective optical properties of the array and shape the temporal profile and statistics of the emitted light. This presents the opportunity to use atomic arrays to produce directional single photons[58], correlated photons[13], or superradiant lasers[59]. Alternatively, measurement of the emitted light provides a window into the complex evolution of the atomic system, and directional detection may enable heralded production of many-body entangled dark states.

## Methods

**Atom–atom interactions**. We consider $N$ two-level atoms of resonance frequency $\omega_0$ and spontaneous emission rate $\Gamma_0$ in free space at positions $\{\mathbf{r}_i\}$. After tracing out the electromagnetic field using a Born–Markov approximation[26,27], the atomic density matrix $\rho$ evolves as

$$\dot{\rho} = -\frac{i}{\hbar}[\mathcal{H}, \rho] + \sum_{i,j=1}^{N} \frac{\Gamma_{ij}}{2}\left(2\hat{\sigma}_{ge}^j \rho \hat{\sigma}_{eg}^i - \rho\,\hat{\sigma}_{eg}^i \hat{\sigma}_{ge}^j - \hat{\sigma}_{eg}^i \hat{\sigma}_{ge}^j \rho\right), \quad (4)$$

where $\hat{\sigma}_{ge}^i = |g_i\rangle\langle e_i|$ is the atomic coherence operator, $|e_i\rangle$ and $|g_i\rangle$ are the excited and ground states of the $i$th atom, and the Hamiltonian reads

$$\mathcal{H} = \hbar \sum_{i=1}^{N} \omega_0 \hat{\sigma}_{ee}^i + \hbar \sum_{i,j=1}^{N} J^{ij} \hat{\sigma}_{eg}^i \hat{\sigma}_{ge}^j. \quad (5)$$

The coherent and dissipative interaction rates between atoms $i$ and $j$ are given by[12,28]

$$J^{ij} - i\frac{\Gamma^{ij}}{2} = -\frac{\mu_0 \omega_0^2}{\hbar}\,\wp^* \cdot \mathbf{G}_0(\mathbf{r}_i, \mathbf{r}_j, \omega_0) \cdot \wp, \quad (6)$$

where $\wp$ is the dipole matrix element of the atomic transition and $\mathbf{G}_0(\mathbf{r}_i, \mathbf{r}_j, \omega_0)$ is the propagator of the electromagnetic field between atomic positions $\mathbf{r}_i$ and $\mathbf{r}_j$[26,27]

$$\mathbf{G}_0(\mathbf{r}_{ij}, \omega_0) = \frac{e^{ik_0 r_{ij}}}{4\pi k_0^2 r_{ij}^3}\left[(k_0^2 r_{ij}^2 + ik_0 r_{ij} - 1)\mathbb{1} + (-k_0^2 r_{ij}^2 - 3ik_0 r_{ij} + 3)\frac{\mathbf{r}_{ij} \otimes \mathbf{r}_{ij}}{r_{ij}^2}\right], \quad (7)$$

where $\mathbf{r}_{ij} = \mathbf{r}_i - \mathbf{r}_j$ and $r_{ij} = |\mathbf{r}_{ij}|$. The dissipative interactions can be recast in terms of jump operators, $\{\hat{\mathcal{O}}_\nu\}$, found as the $N$ eigenvectors of the matrix $\Gamma$ with elements $\Gamma_{ij}$. The decay rates, $\{\Gamma_\nu\}$, are found as the corresponding eigenvalues. The atomic master equation thus reads

$$\dot{\rho} = -\frac{i}{\hbar}[\mathcal{H}, \rho] + \sum_{\nu=1}^{N} \frac{\Gamma_\nu}{2}\left(2\hat{\mathcal{O}}_\nu \rho \hat{\mathcal{O}}_\nu^\dagger - \rho \hat{\mathcal{O}}_\nu^\dagger \hat{\mathcal{O}}_\nu - \hat{\mathcal{O}}_\nu^\dagger \hat{\mathcal{O}}_\nu \rho\right). \quad (8)$$

The jump operators are generically a superposition of lowering operators and can be expanded as

$$\hat{\mathcal{O}}_\nu = \sum_{i=1}^{N} \alpha_{\nu,i} \hat{\sigma}_{ge}^i, \quad \text{where} \quad \sum_{i=1}^{N} \alpha_{\nu,i}^* \alpha_{\mu,i} = \delta_{\nu\mu} \quad \text{and} \quad \sum_{\nu=1}^{N} \Gamma_\nu |\alpha_{\nu,i}|^2 = \Gamma_0. \quad (9)$$

In the above expression, $\delta_{\mu\nu}$ is the Kronecker delta function and $\alpha_{\nu,i}$ is the spatial profile of the $\nu$ − jump operator. The total photon emission rate is calculated as

$$R = \sum_{\nu=1}^{N} \Gamma_\nu \langle \hat{\mathcal{O}}_\nu^\dagger \hat{\mathcal{O}}_\nu \rangle. \quad (10)$$

**Derivation of the second-order correlation function $g^{(2)}(0)$**. The second-order correlation function is calculated as

$$g^{(2)}(0) = \frac{\sum_{\nu,\mu=1}^{N} \Gamma_\nu \Gamma_\mu \langle \hat{\mathcal{O}}_\nu^\dagger \hat{\mathcal{O}}_\mu^\dagger \hat{\mathcal{O}}_\mu \hat{\mathcal{O}}_\nu \rangle}{\left(\sum_{\nu=1}^{N} \Gamma_\nu \langle \hat{\mathcal{O}}_\nu^\dagger \hat{\mathcal{O}}_\nu \rangle\right)^2}, \quad (11)$$

where the expectation value is taken on the fully excited state $|e\rangle^{\otimes N}$, which is the initial state of the system. Substituting in the form of the operators, as shown in Eq. (9), one finds

$$g^{(2)}(0) = \frac{\sum_{\nu,\mu=1}^{N} \Gamma_\nu \Gamma_\mu \sum_{i,j,l,m=1}^{N} \alpha_{\nu,i}^* \alpha_{\mu,j}^* \alpha_{\mu,l} \alpha_{\nu,m} \langle \hat{\sigma}_{eg}^i \hat{\sigma}_{eg}^j \hat{\sigma}_{ge}^l \hat{\sigma}_{ge}^m \rangle}{\left(\sum_{\nu=1}^{N} \Gamma_\nu \sum_{i,j=1}^{N} \alpha_{\nu,i}^* \alpha_{\nu,j} \langle \hat{\sigma}_{eg}^i \hat{\sigma}_{ge}^j \rangle\right)^2}. \quad (12)$$

On the fully excited state, these expectation values are evaluated as

$$\langle \hat{\sigma}_{eg}^i \hat{\sigma}_{ge}^j \rangle = \delta_{ij}, \quad \langle \hat{\sigma}_{eg}^i \hat{\sigma}_{eg}^j \hat{\sigma}_{ge}^l \hat{\sigma}_{ge}^m \rangle = \left(\delta_{im}\delta_{jl} + \delta_{il}\delta_{jm}\right)\left(1 - \delta_{ij}\right). \quad (13)$$

Therefore,

$$g^{(2)}(0) = \frac{\sum_{\nu,\mu=1}^{N} \Gamma_\nu \Gamma_\mu \left(\sum_{i,j=1}^{N} |\alpha_{\nu,i}|^2 |\alpha_{\mu,j}|^2 + \sum_{i,j=1}^{N} \alpha_{\nu,i}^* \alpha_{\mu,j}^* \alpha_{\mu,i} \alpha_{\nu,j} - 2\sum_{i=1}^{N} |\alpha_{\nu,i}|^2 |\alpha_{\mu,i}|^2\right)}{\left(\sum_{\nu=1}^{N} \Gamma_\nu \sum_{i=1}^{N} |\alpha_{\nu,i}|^2\right)^2}$$

$$= \frac{\sum_{\nu,\mu=1}^{N} \Gamma_\nu \Gamma_\mu \left[\left(\sum_{i=1}^{N} |\alpha_{\nu,i}|^2\right)\left(\sum_{j=1}^{N} |\alpha_{\mu,j}|^2\right) + \left(\sum_{i=1}^{N} \alpha_{\nu,i}^* \alpha_{\mu,i}\right)\left(\sum_{j=1}^{N} \alpha_{\mu,j}^* \alpha_{\nu,j}\right) - 2\sum_{i=1}^{N} |\alpha_{\nu,i}|^2 |\alpha_{\mu,i}|^2\right]}{N^2 \Gamma_0^2}$$

$$= \frac{\sum_{\nu,\mu=1}^{N} \Gamma_\nu \Gamma_\mu \left[1 + \delta_{\nu\mu} - \sum_{i=1}^{N} 2|\alpha_{\nu,i}|^2 |\alpha_{\mu,i}|^2\right]}{N^2 \Gamma_0^2} = \frac{N^2 \Gamma_0^2 + \sum_{\nu=1}^{N} \Gamma_\nu^2 - 2\sum_{i=1}^{N}\left(\sum_{\nu=1}^{N} \Gamma_\nu |\alpha_{\nu,i}|^2\right)\left(\sum_{\mu=1}^{N} \Gamma_\mu |\alpha_{\mu,i}|^2\right)}{N^2 \Gamma_0^2}$$

$$= 1 + \sum_{\nu=1}^{N}\left(\frac{\Gamma_\nu}{N\Gamma_0}\right)^2 - \frac{2}{N} \equiv 1 + \frac{1}{N}\left[\mathrm{Var}\left(\frac{\{\Gamma_\nu\}}{\Gamma_0}\right) - 1\right]. \quad (14)$$

**Derivation of the third-order correlation function $g^{(3)}(0)$.** The third-order correlation function is calculated as

$$g^{(3)}(0) = \frac{\sum_{\nu,\mu,\chi=1}^{N} \Gamma_\nu \Gamma_\mu \Gamma_\chi \left\langle \hat{\mathcal{O}}_\nu^\dagger \hat{\mathcal{O}}_\mu^\dagger \hat{\mathcal{O}}_\chi^\dagger \hat{\mathcal{O}}_\chi \hat{\mathcal{O}}_\mu \hat{\mathcal{O}}_\nu \right\rangle}{\left( \sum_{\nu=1}^{N} \Gamma_\nu \left\langle \hat{\mathcal{O}}_\nu^\dagger \hat{\mathcal{O}}_\nu \right\rangle \right)^3}.$$

$$= \frac{\sum_{\nu,\mu,\chi=1}^{N} \Gamma_\nu \Gamma_\mu \Gamma_\chi \sum_{i,j,l,m,n,p=1}^{N} \alpha_{\nu,i}^* \alpha_{\mu,j}^* \alpha_{\chi,l}^* \alpha_{\chi,m} \alpha_{\mu,n} \alpha_{\nu,p} \left\langle \hat{\sigma}_{eg}^i \hat{\sigma}_{eg}^j \hat{\sigma}_{eg}^l \hat{\sigma}_{ge}^m \hat{\sigma}_{ge}^n \hat{\sigma}_{ge}^p \right\rangle}{\left( \sum_{\nu=1}^{N} \Gamma_\nu \sum_{i,j=1}^{N} \alpha_{\nu,i}^* \alpha_{\nu,j} \left\langle \hat{\sigma}_{eg}^i \hat{\sigma}_{ge}^j \right\rangle \right)^3}$$

(15)

For the fully-excited state, the denominator is $(N\Gamma_0)^3$. For the numerator, the expectation value is

$$\left\langle \hat{\sigma}_{eg}^i \hat{\sigma}_{eg}^j \hat{\sigma}_{eg}^l \hat{\sigma}_{ge}^m \hat{\sigma}_{ge}^n \hat{\sigma}_{ge}^p \right\rangle = \left[ \delta_{ip} \left( \delta_{jn}\delta_{lm} + \delta_{jm}\delta_{ln} \right) + \delta_{in} \left( \delta_{jp}\delta_{lm} + \delta_{jm}\delta_{lp} \right) \right.$$
$$\left. + \delta_{im} \left( \delta_{jp}\delta_{ln} + \delta_{jn}\delta_{lp} \right) \right] \times \left( 1 - \delta_{ij} - \delta_{il} - \delta_{jl} + 2\delta_{ij}\delta_{il} \right).$$

(16)

Using the same relations as above, we calculate the value of $g^{(3)}(0)$ as

$$g^{(3)}(0) = \frac{1}{N^3\Gamma_0^3} \sum_{\nu=1}^{N} \sum_{\mu=1}^{N} \sum_{\chi=1}^{N} \Gamma_\nu \Gamma_\mu \Gamma_\chi \left( 1 + 2\delta_{\nu\mu\chi} + \delta_{\nu\mu} + \delta_{\nu\chi} + \delta_{\mu\chi} + 12 \sum_{i=1}^{N} |\alpha_{\nu,i}|^2 |\alpha_{\mu,i}|^2 |\alpha_{\chi,i}|^2 \right.$$
$$- 2 \sum_{i=1}^{N} |\alpha_{\nu,i}|^2 |\alpha_{\chi,i}|^2 - 2 \sum_{i=1}^{N} |\alpha_{\nu,i}|^2 |\alpha_{\mu,i}|^2 - 2 \sum_{i=1}^{N} |\alpha_{\mu,i}|^2 |\alpha_{\chi,i}|^2$$
$$\left. - 4\delta_{\nu\chi} \sum_{i=1}^{N} |\alpha_{\nu,i}|^2 |\alpha_{\mu,i}|^2 - 4\delta_{\nu\mu} \sum_{i=1}^{N} |\alpha_{\nu,i}|^2 |\alpha_{\chi,i}|^2 - 4\delta_{\mu\chi} \sum_{i=1}^{N} |\alpha_{\nu,i}|^2 |\alpha_{\mu,i}|^2 \right)$$
$$= \frac{1}{N^3\Gamma_0^3} \left( N^3\Gamma_0^3 + 2\sum_{\nu=1}^{N} \Gamma_\nu^3 + 3N\Gamma_0 \sum_{\nu=1}^{N} \Gamma_\nu^2 + 12N\Gamma_0^3 - 6N^2\Gamma_0^3 - 12\Gamma_0 \sum_{\nu=1}^{N} \Gamma_\nu^2 \right)$$
$$= 1 + 2\sum_{\nu=1}^{N} \left( \frac{\Gamma_\nu}{N\Gamma_0} \right)^3 + \left( 3 - \frac{12}{N} \right) \sum_{\nu=1}^{N} \left( \frac{\Gamma_\nu}{N\Gamma_0} \right)^2 + \frac{12}{N^2} - \frac{6}{N}.$$

(17)

## Data availability

All data in this manuscript are available upon reasonable request.

## Code availability

All code used in this manuscript is available upon reasonable request.

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

## Acknowledgements

We are grateful to L. A. Orozco, I. Ferrier-Barbut, A. Browaeys, D. E. Chang, M. Lipson, and E. Sierra for discussions. Research was supported by Programmable Quantum Materials, an Energy Frontier Research Center funded by the U.S. Department of Energy (DOE), Office of Science, Basic Energy Sciences (BES). We acknowledge computing resources from Columbia University's Shared Research Computing Facility project, which is supported by NIH Research Facility Improvement Grant 1G20RR030893-01, and associated funds from the New York State Empire State Development, Division of Science Technology and Innovation (NYSTAR) Contract C090171, both awarded April 15, 2010.

## Author contributions

The numerical analysis was carried out by S.J.M. S.J.M. and A.A.-G. contributed to the development of theoretical ideas and tools, and to the writing of the manuscript.

## Competing interests

The authors declare no competing interests.
