## [Peer Review File · Nature Communications]

REVIEWERS' COMMENTS

Reviewer #1 (Remarks to the Author):

In my view, the authors have resolved all my remarks convincingly. Regarding my main criticism, it remains a fact that the analysis is (also) based on numerical analysis, but in their response and the revised manuscript the authors make a strong argument that this analysis is rather comprehensive and warrants their claims. Similarly, the other remarks were well resolved.

I do not agree with the second reviewer regarding his remark on the "generic phenomenon in line with previous work". Of course most researchers in the field would not object to the idea that superradiance is likely to occur as long as the medium is sufficiently dense. But I am not aware of a quantitative version of this expectation, in particular for ordered arrays. This is what the manuscript provides. I also do not agree to the summary that the analysis of the initial phase of the decay is less relevant since it omits the many interesting aspects throughout the long-time delay. Here, "interesting aspects" translate into "usually too difficult to treat analytically", and it is precisely the idea of the present work to circumvent these severe difficulties and still be able to make strong statements about the dynamics by focusing on the initial evolution.

In my view, the idea of making strong statements about the entire dynamics based on one instance in time is novel in this context and quite intriguing, and will likely spark considerable further research. Overall, I therefore recommend publication of the manuscript.